# Topochemical synthesis of different polymorphs of polymers as a paradigm for tuning properties of polymers

Raja Mohanrao[1,2], Kuntrapakam Hema[1,2] & Kana M. Sureshan [1]✉

Different packing is a mechanism through which nature can produce materials of different properties from the same basic units. There is great interest in constructing different forms of the same polymer by utilising different packing. Common solution-synthesized polymers are amorphous and their post-synthesis crystallization into different topologies is almost impossible. Here we show solid-state polymerization of different reactive polymorphs of a monomer pre-organized in different topologies. Trimorphs of a dipeptide monomer pack in a head-to-tail fashion, placing the azide and alkyne of adjacent monomers in proximity. On heating, these crystals undergo a topochemical azide-alkyne cycloaddition reaction yielding triazole-linked polymer in three different crystalline states; one with antiparallel arrangement of polymer chains, another with parallelly oriented chains, and a third form containing a 1:1 blend of two different conformers aligned in parallel. This approach of exploiting different polymorphs of a monomer for topochemical polymerization to yield polymorphs of polymers is promising for future research.

[1] School of Chemistry, Indian Institute of Science Education and Research Thiruvananthapuram, Kerala 695551, India. [2] These authors contributed equally: Raja Mohanrao, Kuntrapakam Hema. ✉email: kms@iisertvm.ac.in

Nature constructs different materials exhibiting very different properties from a common basic constituent by adopting different packing. For instance, diamond and graphite are different forms of carbon yet their properties pole apart owing to their packing differences. Polymers have revolutionized the way we live by way of augmenting various technological advances[1]. The search for polymers having interesting properties continues. It would be possible to get different forms of the same polymer by imparting different packing in their 3D structures[2–5]. However, the polymers formed in traditional solution-phase synthesis are usually amorphous in nature and are difficult to crystallize[6]. On the other hand, topochemical polymerization offers highly crystalline polymers that are not accessible via traditional solution-phase polymerization[7–31]. Polymorphism, the ability of a solid compound to exist in different crystalline forms, originates from different packing of molecules and there is an ongoing quest for polymorphic forms of crystalline materials in pursuit of attractive properties[32,33]. A plausible way to have access to different forms of polymers having different properties is the topochemical polymerization of different reactive polymorph(s) of a particular monomer[34].

Here, we report topochemical polymerization of three different polymorphs of a monomer to yield three different forms (polymorphs) of the polymer having very different packing and properties. The dipeptide (DP) forms three polymorphs (Fig. 1) and in all the polymorphs, DP molecules adopt head-to-tail arrangement and thereby bring the reactive azide and alkyne groups at proximity for their topochemical azide-alkyne cycloaddition (TAAC) reaction. Upon heating, all the three polymorphs undergo TAAC polymerization to give polymorphs of the polymer. Owing to the difference in their packing, different polymorphs of the polymer exhibit different properties.

## Results and Discussion

**Polymorphs from gel-to-crystal transitions**. We have previously designed the β-sheet forming DP (Fig. 1) terminally decorated with complimentary reacting groups viz. azide and alkyne for its topochemical polymerization via TAAC reaction[35]. In the crystals of DP obtained from a 1:1 mixture of MeOH and toluene (DP-I, rectangular plates, orthorhombic, $P2_12_12_1$, Fig. 2a), as anticipated, the molecules packed in β-sheet arrangement (Fig. 2b) with head-to-tail alignment of molecules in both the directions perpendicular to the H-bonding direction (blue and magenta, Fig. 2c). Also in each of these directions, head-to-tail pre-organized non-covalent chains are arranged antiparallely (adjacent chains run in opposite direction; e.g., two adjacent blue arrows or magenta arrows, Fig. 2c). Upon mild heating, these crystals underwent regiospecific TAAC polymerization forming 1,4-triazolyl linked pseudoprotein, in a single-crystal-to-single-crystal (SCSC) manner. In the resultant polymer crystals, adjacent polymer strands are oriented in opposite directions. Interestingly, we found that DP is an organogelator for various aromatic organic solvents.

Gels being a kinetically trapped metastable state, they can slowly transform to thermodynamically stable crystals[36–39], often to polymorphs that are inconceivable from the normal solution-state crystallization[36,40–44]. Remarkably, the organogels of DP (≥4 wt%) slowly transformed to crystals over a period of time when left undisturbed. While the toluene (3–4 h) and o-xylene (12–15 h) gels transformed to crystals spontaneously at room temperature, gels in m-xylene, benzene, chlorobenzene got converted to crystals when refrigerated (0–5 °C). The complete transition of the gels to crystals occurred in 5–6 days. The morphologies of the crystals obtained from gels were different from the rectangular plate morphology of crystals obtained from

the solution-state crystallization (Fig. 1b). The toluene gel (Fig. 1c) slowly transformed to several large swallow-tail twinned crystals (m.p. 129 °C). But all other gels yielded flower-like clusters (Fig. 1c) consisting of several thin needle-shaped crystals (m.p. 125–129 °C) originating from a common nucleation point. DSC thermograms of crystals from solution and different gels clearly showed difference in their melting temperatures (Fig. 1d). From the difference in their morphologies and DSC thermograms, it is apparent that they are different polymorphs of DP. Powder X-ray diffraction (PXRD) profiles of these crystals also confirmed that they are different polymorphs (Fig. 1e). It is interesting that such minute difference in solvent leads to different polymorphs[45]. Instances of gel-to-crystal transitions resulting in interesting structural forms are recently reported[46–49] and this prompted us for a thorough scrutiny of these gel-derived crystals. Accordingly, we have carried out the single-crystal X-ray diffraction (SCXRD) analyses of these gel-derived crystals.

The single-crystal analyses of the crystals obtained from gels confirmed that these are two polymorphs of DP (Fig. 2). While the rectangular crystals (Fig. 2a–c) from solution (DP-I) adopted orthorhombic $P2_12_12_1$ space group, the gel-derived polymorphs adopted monoclinic $P2_1$ space group. The toluene gel-derived twinned polymorph (DP-II, Fig. 2d–f and Supplementary Fig. 1) has three symmetry-independent molecules in the asymmetric unit ($Z' = 3$) (represented as A–C in purple, green, and black colors, respectively) and the clustered crystal derived from other gels (DP-III) has one molecule in the asymmetric unit ($Z' = 1$, Fig. 2g–i and Supplementary Fig. 1; for full crystal data of crystals derived from various gels see Supplementary Tables 1–4). All the three polymorphs adopt β-sheet arrangement of the dipeptide molecules (Fig. 2b, e, h). In all these polymorphs, in the plane perpendicular to the hydrogen bonding direction (i.e., β-sheet forming direction), the dipeptide molecules are arranged in head-to-tail fashion, thus disposing the complementary reactive groups viz. alkyne and azide proximal to each other (Fig. 2c, f, i). However, the azide and alkyne are not in a ready-to-react parallel orientation in any of these polymorphs. Nevertheless, the flexibility of the azide and alkyne for rotation and the availability of rooms around the reaction cavity allow these polymorphs to attain reactive orientation.

A comparison of the crystal packing shows stark contrast between DP-I and other two polymorphs. In the case of polymorph DP-II, the three symmetry-independent molecules A–C are arranged successively in the β-sheet (Fig. 2e). Adjacent β-sheets are arranged in opposite directions and this arrangement makes head-to-tail arrangement of molecules in the perpendicular direction (b-direction) with proximally placed azide and alkyne such that alternatively arranged conformers A and B make a head-to-tail arranged chain (azide of conformer A is proximal to the alkyne of conformer B and vice versa) and conformers C make another self-sorted head-to-tail chain (Fig. 2f). Though azide and alkyne are not in a ready-to-react parallel orientation, by the rotation of azide by 50–60° around C–N bond, reactive orientation to yield 1,4-linked triazole can be attained. This kind of alignment may result in two different conformers of polymer within a single-crystal upon TAAC reaction. Molecules in polymorph DP-III also arrange in parallel β-sheets (Fig. 2h) and the molecules adopted head-to-tail arrangement in the perpendicular direction (along b-direction, Fig. 2i). As in the case of other two polymorphs, the reactive groups in DP-III are also at proximity but not in an ideal parallel arrangement for their topochemical reaction. However, rotation of the azide around C–N bond by 57° would result in a reactive conformation to yield 1,4-triazolyl linked polymer. As DP-II and DP-III have similar azide-alkyne proximity as in DP-I, which underwent TAAC

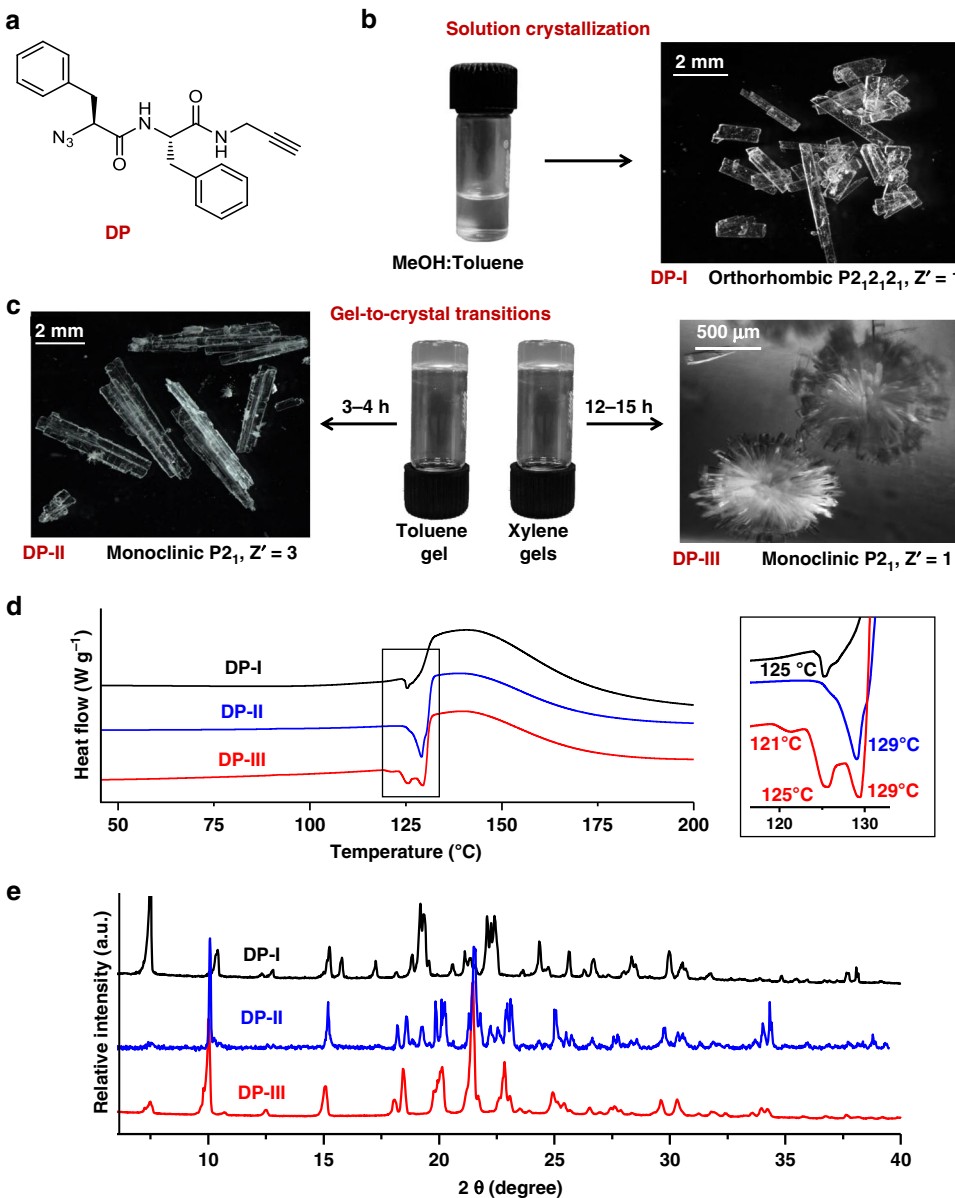

**Fig. 1 Polymorphs of the dipeptide (DP). a** Chemical structure of the dipeptide (DP). **b** Crystals of the dipeptide (DP-I) obtained from solution (MeOH:Toluene). Scale bar = 2 mm. **c** Crystals of the dipeptide obtained from gels; toluene gel (DP-II, scale bar = 2 mm) and xylene gel (DP-III, scale bar = 500 μm). **d** DSC thermograms of the trimorphs of DP. **e** PXRD patterns of the trimorphs of DP. Source data are provided as a Source Data file for Fig. 1d, e.

polymerization under mild conditions, we anticipated that these two polymorphs also would undergo TAAC polymerization. Employing these polymorphs for the topochemical reaction would be interesting since the difference in the packing of monomer molecules could be exploited for the synthesis of structurally and functionally different forms of the same polymer.

**Topochemical polymerization of the polymorphs.** In order to see the topochemical reactivities of polymorphs DP-II and DP-III, we have heated these crystals at 60 °C (Supplementary Methods) and monitored the progress of their reaction by time-dependent IR spectroscopy and $^1$H nuclear magnetic resonance (NMR) spectroscopy. For this, we withdrew small fraction of sample at different times and recorded their IR spectra (solid state using KBr pellet) and $^1$H NMR spectra after dissolving in DMSO-$d_6$. The

gradual diminishing of the signal due to azide (2105 cm$^{-1}$) in the time-dependent IR spectroscopy suggested the gradual consumption of monomer with time (Supplementary Fig. 2). Similarly, time-dependent DSC analysis also suggested the gradual consumption of monomer with time (Supplementary Fig. 2)[27]. The gradual consumption of the dipeptide and consequent formation of oligomers/polymers was also adjudged based on the disappearance of the signals due to monomer and formation of signals corresponding to the triazole-linked oligomers/polymers in the $^1$H NMR spectra (Fig. 3a, b). At all stages of the reaction, only one kind of triazolyl proton signal (at 7.87 ppm) was present and this suggested the occurrence of regiospecific TAAC reactions in both the polymorphs. While DP-II got completely converted to the polymer PP-II in 10 days, DP-III took 12 days for its polymerization to PP-III. The difference in reactivity might have originated from the different packing in these polymorphs. The

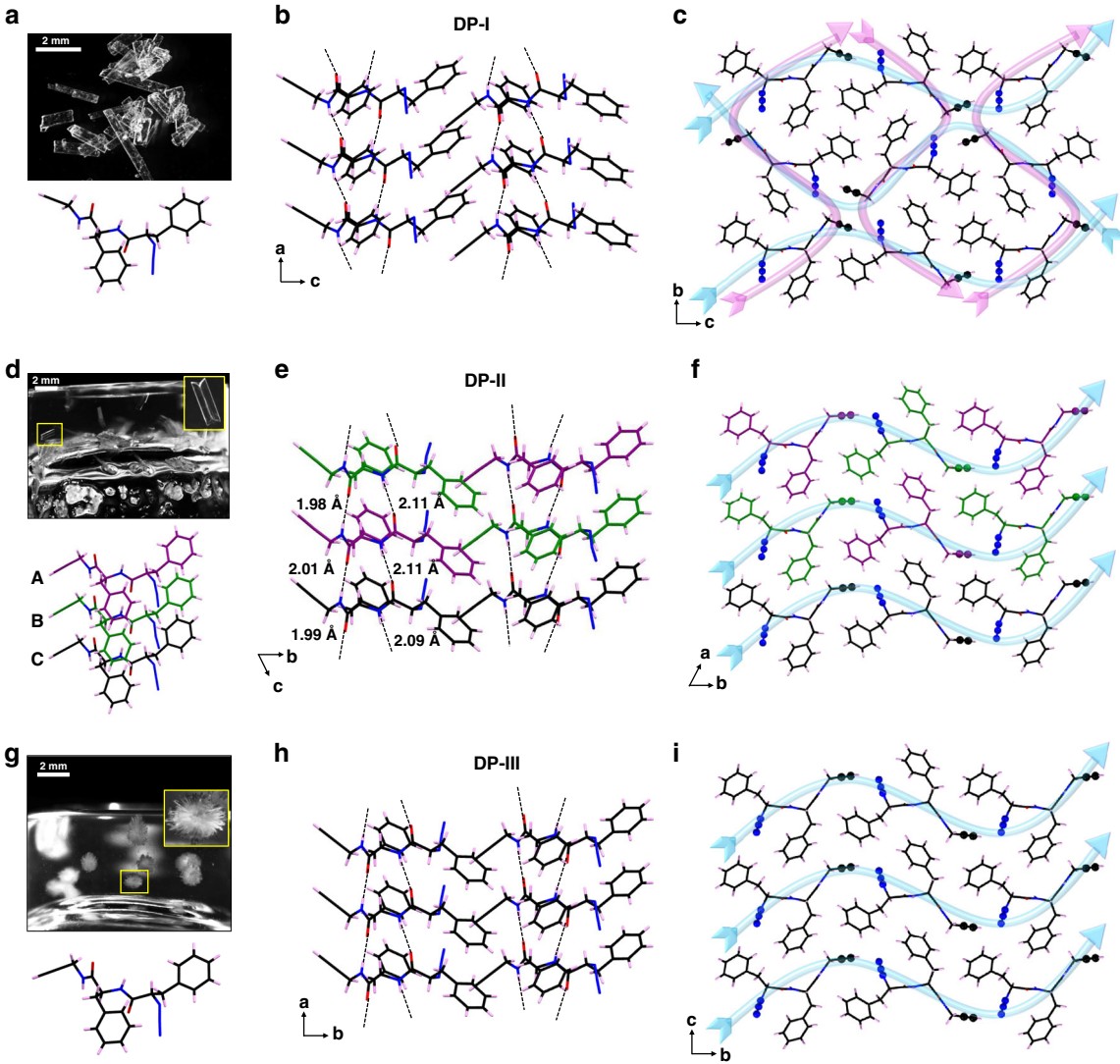

**Fig. 2 Crystal structures of the three polymorphs of the dipeptide DP. a** Crystals of DP-I obtained from solution (MeOH/toluene). Scale bar = 2 mm. **b** β-sheet alignment in form DP-I. **c** Head-to-tail alignment of molecules in both b- and c-directions and two pairs of reactive groups entrapped in a cavity-like environment in DP-I. **d** Formation of twinned crystals (DP-II) in toluene gel with three symmetry-independent molecules in the asymmetric unit. Scale bar = 2 mm. **e** β-sheet alignment in DP-II showing alternative alignment of the conformers **A–C** in the direction of hydrogen bonding **f** Head-to-tail alignment of molecules in DP-II in b-direction; while **A** and **B** are aligned alternatively in the reaction direction, **C** adopted self-sorted alignment. **g** Formation of flower-like microstructures (DP-III) inside o-xylene gel. Scale bar = 2 mm. **h** β-sheet alignment in form DP-III. **i** Head-to-tail alignment of molecules in DP-III in b-direction.

reactivities of both polymorphs exhibited sigmoidal kinetics, suggesting that both the polymorphs reacted topochemically (Fig. 3c, d). [1]H NMR spectra of isotropic solutions of both the pseudoproteins PP-II and PP-III were identical with that of previously reported 1,4-triazolyl linked polymer, PP-I obtained by the SCSC polymerization of DP-I. Thus both polymorphs DP-II and DP-III underwent TAAC polymerization akin to polymorph DP-I to yield polymorphs of 1,4-triazole-linked polymer. The MALDI-TOF analysis of the fully reacted DP-II and DP-III showed the presence of oligomers up to 28-mers and 21-mers, respectively but in the case of DP-I, oligomers of up to 35-mers were observed (Supplementary Fig. 3). Gel permeation chromatography analysis of the pseudoproteins PP-I–PP-III revealed the presence of polymers with an average Mw of 56,313, 41,118, and 39,519 g mol$^{-1}$, respectively (Supplementary Fig. 4).

We have monitored the solid-state reaction of polymorphs to polymers by using time-dependent PXRD studies. For this, 15 mg

of each of these polymorphs were heated at 60 °C and recorded their PXRD profile at different times. Both the polymorphs reacted without losing their crystallinity. In the case of polymorph DP-II, at every stage of the reaction sharp peaks were observed (Fig. 3e). While some sharp peaks of the monomer phase diminished, some shifted and some additional peaks appeared as the reaction progressed. In the case of DP-III sharp peaks were observed during the initial stages of the reaction and at the end of the reaction broadening was observed for some of the peaks (Fig. 3f). As PXRD analysis suggested the conservation of crystallinity, we attempted to solve the crystal structures of fully reacted crystals by SCXRD analysis. The crystals of PP-II diffracted well and were suitable for SCXRD analysis (Supplementary Fig. 5). However, the crystals of PP-III showed weak diffraction suggestive of loss of quality of the single-crystals after the complete reaction (Supplementary Fig. 6). This is in agreement with the broadening observed in its PXRD profile.

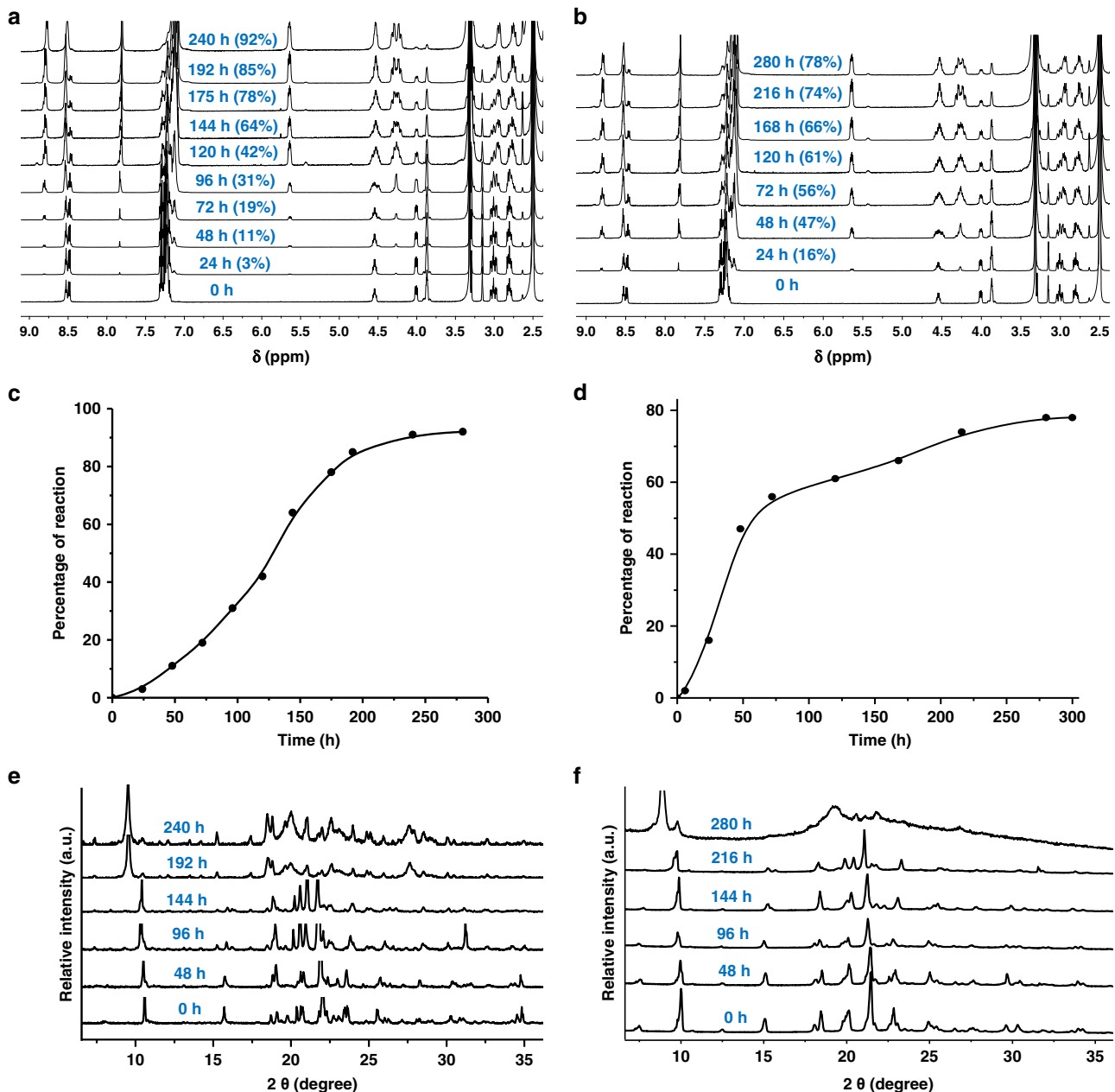

**Fig. 3 TAAC reactions of the gel-derived polymorphs. a, b** Time-dependent $^1$H NMR (DMSO-d$_6$) of DP-II and DP-III, respectively. **c, d** Kinetics graph of DP-II and DP-III, respectively. **e, f** Time-dependent PXRD spectra of DP-II and DP-III, respectively. Source data are provided as a Source Data file for Fig. 3c–f.

SCXRD analysis of PP-II revealed that the DP-II underwent a SCSC polymerization (Fig. 4) and the polymer preserved the monoclinic P2$_1$ space group as in the DP-II. The pseudoprotein has conserved the β-sheet alignment like in dipeptide DP-II (Fig. 4e). The TAAC reaction in PP-II happened along the crystallographic b-direction as predicted (Fig. 4f). As expected based on the monomer packing (vide supra), though DP-II has three symmetry-independent molecules in its asymmetric unit (Z′ = 3), only two symmetry-independent molecules are present in the asymmetric unit (Z′ = 2, Fig. 4d, **A** and **B** in black and green colors, respectively) in PP-II. While the three symmetry-independent molecules in the DP-II show very slight differences in their conformations, in the corresponding pseudoprotein PP-II the two symmetry-independent molecules have adopted completely dissimilar conformations with large reorientation of the

phenyl rings (Supplementary Figs. 7 and 8). Having two conformationally different polymer chains in the same crystal is an interesting phenomenon. It is also interesting that the PP-II maintained its crystallinity despite the extensive motion of phenyl groups in the crystal lattice.

The packing of the polymer in PP-II was entirely different from that in PP-I (Fig. 4a–c). DP-I polymerizes along "b" direction (magenta) and produce polymers that are packed antiparallely along "c" direction. Both DP-II and DP-III polymerize along "b" direction (blue) to form polymers that are packed parallelly. This difference in packing also leads to different properties. In PP-I, the TAAC reaction led to the generation of channels that were spontaneously occupied by water molecules, which can be removed reversibly. The reversible water sorption makes it a material for plausible desiccant and

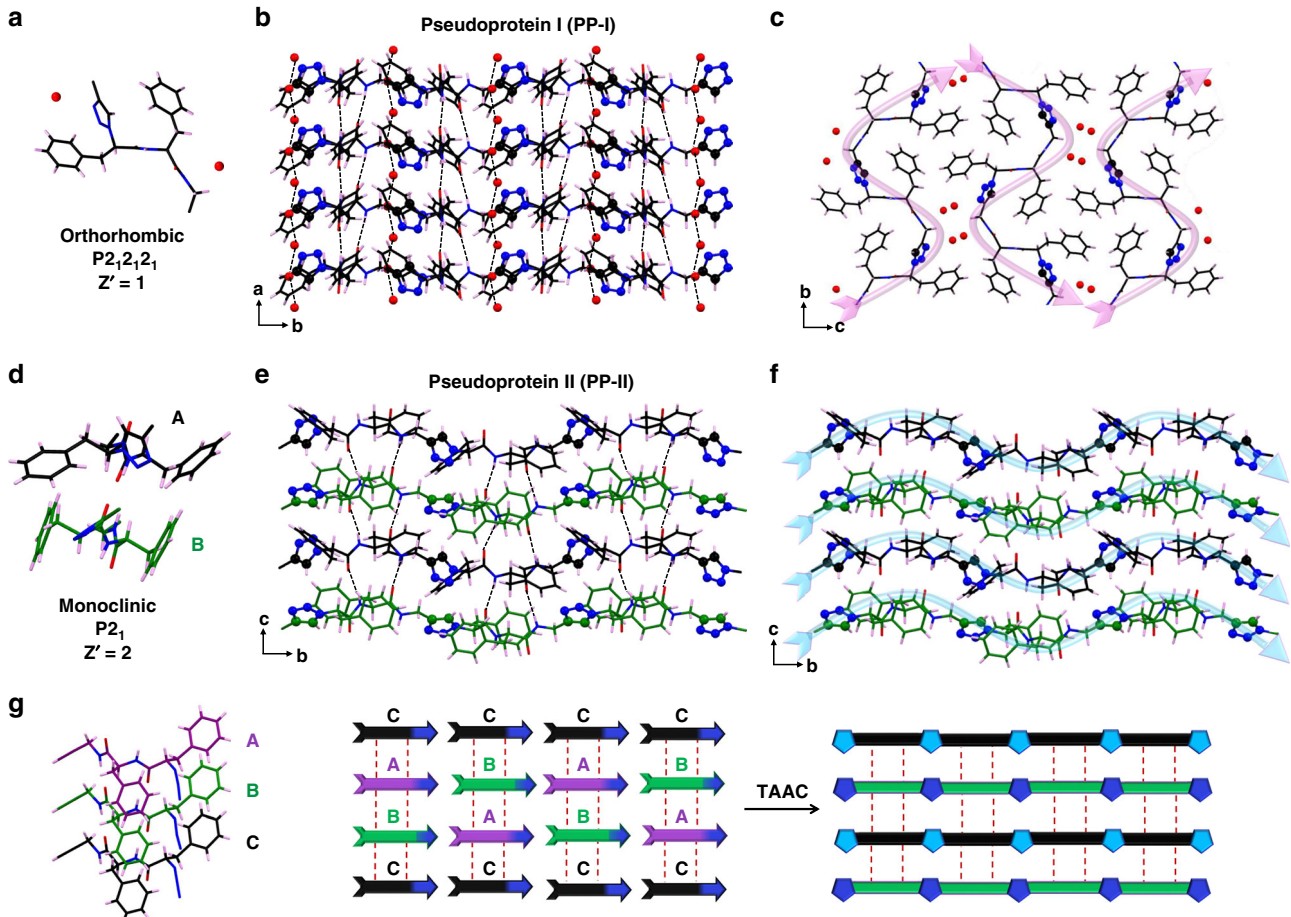

**Fig. 4 Structures of the pseudoproteins. a** Crystal structure of PP-I showing the presence of two water molecules in the asymmetric unit. **b** β-sheet hydrogen bonding in PP-I and one-dimensional water wire. **c** Pseudoprotein packing in PP-I showing water molecules in the channels. **d** Crystal structure of PP-II showing two symmetry-independent molecules in the asymmetric unit. **e** β-sheet hydrogen bonding in the PP-II. **f** Two dissimilar polymer chains in the PP-II with their phenyl rings placed between the β-strands. **g** Schematic showing the TAAC reaction in DP-II leading to two conformationally dissimilar polymer chains in PP-II.

water-harvesting applications. On the other hand, polymers PP-II and PP-III do not absorb water. The polymers also showed different thermal stabilities as evident from the thermogravimetric analysis. While the PP-I and PP-II showed a decomposition temperature 300 °C, polymer PP-III decomposed at 315 °C (Fig. 5a). Similarly, the density of polymer PP-I (anhydrous form) is 1.231 g cc$^{-1}$ and that of PP-II is 1.217 g cc$^{-1}$. The solubilities of the three polymers in different solvents also varied greatly (Supplementary Table 5). For instance, while the PP-I was insoluble in DMSO, other two forms were soluble; PP-II and PP-III showed solubilities of 0.4 and 0.32 g mL$^{-1}$, respectively. The presence of π–π stacking in the PP-I and its absence in other two forms may be responsible for the difference in solubilities. The difference in packing also had an influence in the mechanical behavior of the crystal after polymerizations. All the three polymorphs had undergone cracking along with the reaction (Fig. 5b). Large movement of the groups in the crystal lattice and changes in their volumes upon TAAC reaction would have caused cracking[50] in the crystals. While the DP-I showed extensive cracking, DP-II cracked to a much lower extent, and the DP-III cracked to an intermediate level. The foregoing discussions clearly highlight the importance of exploiting polymorphs for topochemical polymerization to achieve structurally and functionally different polymers.

In summary, we have exploited different polymorphs of the same monomer for topochemical polymerization to give structurally and functionally diverse polymers. Various gels of DP having azide and alkyne at its termini undergo gel-to-crystal transitions to form two polymorphs, which are different from the solution-derived crystal. In all the three polymorphs, the molecules adopt β-sheet packing, which makes molecules to adopt head-to-tail arrangement with proximally placed azide and alkyne in a direction perpendicular to the β-sheet direction. Up on mild heating, all the three polymorphs undergo TAAC polymerization to yield 1,4-triazolyl-linked pseudoprotein with different packing, dictated by the packing in the corresponding polymorph of the monomer. One of the gel-derived polymorphs undergo a SCSC polymerization as in the case of solution-derived polymorph but gives a structurally different polymer. All the three structurally different polymers show different properties, such as density, thermal stability, solubility, and water sorption. Our results suggest that finding different polymorphs of polymerizable monomer and their exploitation in topochemical polymerization may provide different forms of polymers having attractive properties. Though we have used gel-to-crystal transitions for obtaining different polymorphs, the scope of the present study is broad as any crystallization technique can be employed to achieve polymorphs of a monomer. With the recent

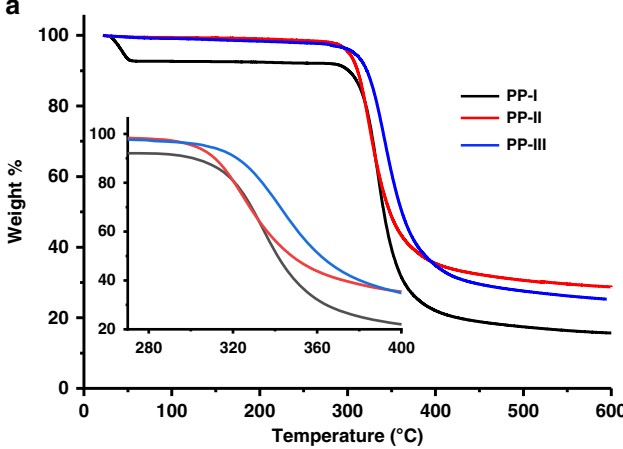

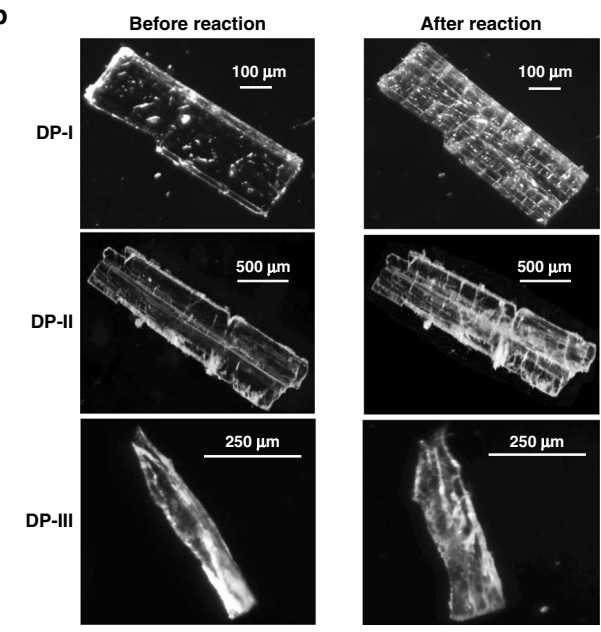

**Fig. 5 Thermal stabilities of the pseudoproteins and the morphology changes due to TAAC reaction. a** TGA comparison of the pseudoproteins PP-I (black), PP-II (red), and PP-III (blue). **b** Microscopic images showing cracking of crystals of all three polymorphs DP-I (scale bar = 100 μm), DP-II (scale bar = 500 μm), and DP-III (scale bar = 250 μm) due to TAAC reaction. Source data are provided as a Source Data file for Fig. 5a.

advancement in the methods of preparing polymorphs, this approach would be practical.

### Data availability

The X-ray crystallographic data have been deposited at the Cambridge Crystallographic Data Centre (CCDC) with the deposition numbers 1914914–1914919. These data can be obtained from The Cambridge Crystallographic Data Centre via www.ccdc.cam.ac.uk/data_request/cif. The source data underlying Figs. 1d, e, 3c–f, 5a, Supplementary Figs. 2a–d, 3a–c are provided as a Source Data file.

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

## Acknowledgements

K.M.S. thanks Department of Science and Technology, India for a SwarnaJayant Fellowship and Science and Engineering Research Board for a research grant (CRG/2018/000577).

## Author contributions

K.M.S. conceived the idea. R.M. and K.H. performed the experimental work and contributed equally. All authors analyzed the results and contributed to the writing of the paper.

## Competing interests

The authors declare no competing interests.
