## [Peer Review File · Nature Communications]

Reviewers' comments:

Reviewer #1 (Remarks to the Author):

The author have exploited different polymorphs of the monomer by gel to solid transitions and subsequent topochemical polymerization in order to demonstrate crystalline polymer. The crystalline structures were specifically identified based on the PXRD patterns. However, the reviewer doesn't understand why the author used the transition from gel to crystal.

1. Different polymorphs of the DP molecule were obtained by gel-to-crystalline transition and also authors used the various solvents. Even though author investigated the crystalline structures, there is no explanation why they have different structures. Is it from the evaporation velocity of solvent or difference of physical interaction between solvents and DP?

2. For the crystalline structure, various methods can be adopted such as solution to solid transition, re-precipitation or thermo-dependent transition. Why author use the gel to solid transition?

3. The reviewer has known that azide-alkyne cycloaddition is occur under copper catalysis. It seems that the catalyst could affect the crystalline structure.

4. In figure 1C, why is DP transformed to crystals more slowly in o-xylene than in toluene? Does the morphology of the crystals obtained from gels affects the transformation time?

5. In figure 5B, the authors suggested that cracking occurred in all three polymorphs after reaction, especially in DP-I and DP-III. However, it is difficult to justify the extent of cracking and it is necessary to provide evidences in more detail.

Reviewer #2 (Remarks to the Author):

I was quite impressed with the author's first reported topochemical polymerization (DP-I) as reported last year in *Angewandte Chemie*. This follow up report extends that work in a very interesting direction. The series of three polymorphic polymers makes the the work much more significant.

The work all looks to be of very high quality. The crystallography looks sound as does the accompanying thermal and spectroscopic data.

I do have a question about the solubility of the polymers. DMSO NMR studies are reported so there must be some solubility. How soluble are they in other solvents?

Since the authors could "easily" extend this work to other dipetide derived molecules, the have the potential for extending this work in many directions. Good luck.

Nice article I recommend publication with great enthusiasm.

Reviewer #3 (Remarks to the Author):

In this article the authors demonstrate the formation of three different crystalline forms of polymers by means of an innovative approach. They have used the reliable strategy of forming

beta sheets to prearrange molecules with favourable topochemical alignment of photoactive groups. The crystals were then subjected to mild heating in order to bring about polymerisation. Remarkably, the polymerisation occurs in single-crystal to single-crystal fashion, allowing them to determine the crystal structures by means of X-ray diffraction. Polymerisation of one of the crystal forms was recently reported by the authors. They have now extended this work significantly by obtaining two additional polymorphs by slowly allowing organogels of the compound to crystallise. Polymorphism was controlled by judicious choice of solvent and the temperature of crystallisation. Slow crystallisation of gels is a technique known to produce polymorphs not obtainable from simple solution growth. The as-prepared crystals were analysed using single-crystal diffraction, and then the compounds were polymerised using heat. The resulting crystals of polymers were then also elucidated crystallographically. In this way the authors formed three different polymorphs of the same polymers. Interestingly, despite not satisfying the proximity criteria for topochemical reactivity (i.e. parallel alignment), the reactions proceeded anyway. The authors reasonably suggest that this is likely due to structural and conformational flexibility. In my view this report represents a significant advance over previous work in this field. These results will stimulate further investigation of the effects of polymorphism on topochemical reactions, thus producing new materials with desirable properties. The work has been carried out with great care and competence and the characterisations fully support the claims. I recommend that the manuscript be accepted as is (while there are some minor oddities in the grammar, I am sure that these can be corrected by the appropriate journal staff).

Our Response to Reviewer Comments

Reviewer #1

The author have exploited different polymorphs of the monomer by gel to solid transitions and subsequent topochemical polymerization in order to demonstrate crystalline polymer. The crystalline structures were specifically identified based on the PXRD patterns. However, the reviewer doesn't understand why the author used the transition from gel to crystal.

Our Response: The key-step is to get different polymorphs of the monomer. Crystallization from different solvent or under different conditions can sometime give different polymorphs. Sometimes gelators undergo spontaneous gel-to-crystal transition and is another way to get novel polymorphs of a molecule. We could not obtain different polymorphs by direct crystallization from solutions. But the monomer formed gel with various solvents and exhibited spontaneous gel-to-crystal transitions giving different polymorphs.

1. Different polymorphs of the DP molecule were obtained by gel-to-crystalline transition and also authors used the various solvents. Even though author investigated the crystalline structures, there is no explanation why they have different structures. Is it from the evaporation velocity of solvent or difference of physical interaction between solvents and DP?

Our Response: Several factors such as temperature, polarity of solvent, interactions between solvent and molecule, concentration of the solution, medium of crystallization, rate of evaporation etc play roles in stabilizing a particular polymorph. Even mild changes in these crystallization conditions could result in a new polymorph and in that sense rate of solvent evaporation, interaction of solvent with solute can also contribute to polymorphism. However, it is unrealistic to pinpoint a specific factor as the sole contributor in stabilizing a polymorph of a molecule.

2. For the crystalline structure, various methods can be adopted such as solution to solid transition, re-precipitation or thermo-dependent transition. Why author use the gel to solid transition?

Our Response: We have presented here the proof of concept of achieving polymorphs of a polymer by topochemical polymerization of polymorphs of the monomer. As the referee correctly mentioned, any crystallization method could be adopted for achieving polymorphs of the monomer as long as it gives different polymorphs. Gel-to-crystal transition is a method of contemporary interest to achieve novel polymorphs of a molecule and this is the method that gave us different polymorphs. In the conclusion, we have added a statement mentioning the broader scope of our work in view of various methods of crystallization that can be adopted for obtaining polymorphs of the monomer.

3. The reviewer has known that azide-alkyne cycloaddition is occur under copper catalysis. It seems that the catalyst could affect the crystalline structure.

Our Response: The copper catalysis for azide-alkyne cycloaddition is relevant in solution-state (click) reaction. The present study deals with the topochemical azide-alkyne cycloaddition reaction that happens in the crystals (controlled by the molecular alignment in the crystal lattice) without the need of catalysts, solvents or any other additive. Hence, there is no catalyst in our reaction.

4. In figure 1C, why is DP transformed to crystals more slowly in o-xylene than in toluene? Does the morphology of the crystals obtained from gels affects the transformation time?

Our Response: It is very similar to crystallization from two different solutions; from some solvent crystallization is fast and from some solvent, crystallization is very slow. Even from the same solvent under same conditions, the crystallization speed can be different. The slow gel-to-crystal transition in o-xylene gel compared to that in toluene gel could be due to several factors such as difference in the molecular interactions with the solvent, rate of solvent evaporation or saturation, minor difference in polarity of the solvents, initial nucleation etc. Hence it is difficult to pinpoint a specific factor that control the transition time.

5. In figure 5B, the authors suggested that cracking occurred in all three polymorphs after reaction, especially in DP-I and DP-III. However, it is difficult to justify the extent of cracking and it is necessary to provide evidences in more detail.

Our Response: This is an experimental observation. We have provided the photographs of the crystal before and after reaction as evidence for cracking. Now for more clarity, we have kept a better photograph. We thank this referee for his/her time in carefully evaluating our manuscript.

Reviewer #2 :

I was quite impressed with the author's first reported topochemical polymerization (DP-I) as reported last year in *Angewandte Chemie*. This follow up report extends that work in a very interesting direction. The series of three polymorphic polymers makes the work much more significant. The work all looks to be of very high quality. The crystallography looks sound as does the accompanying thermal and spectroscopic data.

I do have a question about the solubility of the polymers. DMSO NMR studies are reported so there must be some solubility. How soluble are they in other solvents?

Since the authors could easily extend this work to other dipeptide derived molecules, they have the potential for extending this work in many directions. Good luck. Nice article I recommend publication with great enthusiasm.

Our Response: We thank this referee for these very positive comments of appreciation and recommending acceptance. We have now checked the solubility in common solvents. The solubility data has been added in the Supplementary Information.

REVIEWERS' COMMENTS:

Reviewer #2 (Remarks to the Author):

I have read the review comments and the authors' replies. I believe that all significant points have been answered. Minor revisions to the manuscript have been completed and the paper is now ready for publication.

Overall this is a nice contribution and I look forward to seeing the authors' future work.